# Differential Emodepside Efficacy in Drug-Resistant and Drug-Susceptible *Ancylostoma caninum* Highlights Variability in Potassium Channel Activity

**DOI:** 10.3390/tropicalmed10070181

**Published:** 2025-06-27

**Authors:** Catherine A. Jackson, Elise L. McKean, John M. Hawdon

**Affiliations:** 1Department of Microbiology, Immunology, and Tropical Medicine, The George Washington University, Washington, DC 20052, USA; elisemckean@gwu.edu (E.L.M.); jhawdon@gwu.edu (J.M.H.); 2Department of Biological Sciences, The George Washington University, Washington, DC 20052, USA

**Keywords:** *Ancylostoma caninum*, hookworm infection, multidrug resistance, emodepside, SLO-1 channel, BK channel, L3

## Abstract

Multi-anthelmintic resistance in hookworms poses a significant challenge to both human and veterinary health, underscoring the need for novel treatment strategies. In this study, we evaluated the in vitro efficacy of three anthelmintics—pyrantel, ivermectin, and emodepside—against L3 larvae of drug-susceptible (WMD) and triple-anthelmintic-resistant (BCR) isolates of *Ancylostoma caninum*. While pyrantel was largely ineffective and ivermectin induced high mortality in both isolates, emodepside displayed a surprising trend: the drug-resistant BCR isolate was more susceptible than the drug-susceptible WMD isolate. To explore the underlying mechanism, we performed survival assays in the presence of penitrem A, a BK channel (SLO-1) inhibitor. The addition of penitrem A reversed the enhanced emodepside sensitivity in BCR, implicating elevated basal expression of SLO-1 channels as a potential factor. These findings suggest that emodepside, via its action on SLO-1, may offer a promising therapeutic avenue to combat multidrug-resistant hookworm infections.

## 1. Introduction

Parasitic helminths are a global public health concern, primarily affecting tropical and subtropical regions where socio-economic challenges often hinder access to effective and novel treatments. Infections with intestinal hookworms, including *Necator americanus* and *Ancylostoma duodenale*, can cause iron deficiency anemia and protein malnutrition, and account for the loss of approximately 5.2 million disability-adjusted life years annually [1]. The canid specialist hookworm, *Ancylostoma caninum,* is the most prevalent intestinal parasite of dogs and can cause skin infections in humans [2]. Since becoming widely available in the 1960s, a broad range of anthelmintic drugs have facilitated the effective treatment of intestinal helminths across different stages of the parasite life cycle. Despite the success of drugs like benzimidazoles (e.g., albendazole), macrocyclic lactones (e.g., ivermectin), and tetrahydropyrimidines (e.g., pyrantel), widespread and sometimes improper use of these drugs has accelerated the selection and spread of multi-anthelmintic drug-resistant (MADR) canine hookworm isolates [2,3,4,5]. In the canid hookworm *A. caninum*, double-resistant [6] and triple-resistant [7] isolates have been characterized. While *A. caninum* is unable to establish a gastrointestinal infection in humans, dermal contact with the infective stage larvae may cause cutaneous *larva migrans*, a highly irritating skin infection [8,9]. Currently, the drugs used to treat this condition in humans are the same as those to which resistance has developed in MADR *A. caninum*. As such, the emergence of multidrug resistance has presented substantial challenges in both human and veterinary medicine, complicating treatment and reducing the availability of effective therapies.

The natural rise and spread of MADR *A. caninum* has been confirmed across the continental United States and Canada, with unverified but sensible reasoning that a similar pattern may be transpiring in other areas where hookworms are endemic [10], underscoring the need for novel treatment approaches. Emodepside, a semi-synthetic derivative of the cyclo-octadepsipeptide PF1022A, has emerged as a promising candidate due to its unique mode of action. Originally discovered from the endophytic fungus *Rosellina* sp. [11,12], emodepside targets neuromuscular pathways that are not affected by traditional anthelmintics, including binding to latrophilin-like receptors and activating SLO-1 potassium channels [13]. Orthologs for the genes encoding these molecules in *A. caninum*, *Haemonchus contortus*, and *Cooperia oncophora* were identified and expressed in emodepside-resistant *Caenorhabditis elegans*, elucidating the specific necessity of SLO-1 channels for emodepside to be properly active [14]. The influence of the drug on these pathways induces paralysis, impacts locomotion, and ultimately causes death in nematodes.

In vitro assays have demonstrated the effectivity of emodepside against *Trichuris muris*, *A. ceylanicum*, *N. americanus*, *Heligmosomoides polygyrus*, and *Strongyloides ratti*, and these results were confirmed in vivo for *T. muris*, *N. americanus*, and *A. ceylanicum* [15]. Additionally, the drug has been shown to be effective against parasites with significant resistance to benzimidazoles and macrocyclic lactones. Comparative studies show that emodepside is at least 50% more effective than drugs from other classes in reducing the fecal egg count of dogs infected with MADR *A. caninum* [16]. In a study focused on a population of humans with hookworm infections, emodepside demonstrated superior therapeutic effects compared to albendazole [17]. In the United States, where MADR *A. caninum* is confirmed to be most prevalent, emodepside is approved for use as a topical formulation for cats, but not for dogs. The robust efficacy profile indicates that the mechanism of action for emodepside could potentially bypass common resistance pathways, highlighting its value as a tool for managing MADR hookworm infections.

In this study, we aim to evaluate the in vitro efficacy of emodepside compared to two commonly used anthelmintics, pyrantel and ivermectin, against infective larvae of drug-resistant and susceptible isolates of the canine hookworm *A. caninum*. By comparing these drugs, we seek to highlight the potential of emodepside as a versatile treatment option that may address multidrug resistance and offer cross-species efficacy.

## 2. Materials and Methods

### 2.1. Parasite Isolates Source and Maintenance

The wild-type (WMD) and triple-anthelmintic-resistant (BCR) isolates of *A. caninum* used in this study have been previously described [6,7]. BCR exhibits resistance to thiabendazole, pyrantel, ivermectin, and moxidectin. Both isolates were maintained in beagles according to established protocols [18,19].

### 2.2. Drug Selection and Preparation

Ivermectin, emodepside, pyrantel pamoate, and dimethyl sulfoxide (DMSO) were sourced from Sigma-Aldrich (St. Louis, MO, USA). Penitrem A was purchased from Cayman Chemical Company (Ann Arbor, MI, USA).

#### 2.2.1. Selection of Concentrations

To address the specific challenges associated with targeting *A. caninum* L3 compared to the adult stage, drug concentrations were selected based on reported maximum plasma concentrations (C_max_) from the literature.

Ivermectin: The typical therapeutic dose (150–200 mg/kg) yields a C_max_ of 0.038–0.046 µg/mL after a single oral administration [20].

Pyrantel: Administered at 100 mg/kg, pyrantel pamoate achieves a C_max_ of 0.11 µg/mL [21]. Despite this dose being 10–20 times greater than the standard dose [22], it was selected due to limited data on C_max_ at typical doses and the poor intestinal absorption of pyrantel [20].

Emodepside: Although determination of a standard dose for oral administration of emodepside is ongoing, 1 mg/kg has proven to be effective through oral administration of the feline topical solution and of the canine modified-release tablet [23]. The aforementioned study found significant discrepancy in C_max_ values between oral and topical administration. Thus, to maintain consistency in the route of administration for all three drugs, preliminary data suggesting efficacy at 1–5 mg when administered orally [24] was used to determine the concentration to be tested in vitro. At a median dose of 2.5 mg, emodepside achieves a C_max_ of 37.6 ng/mL [25].

Given that these C_max_ values correspond to doses that affect the more vulnerable adult stage, each was multiplied by a factor of 1000 to simulate concentrations that could target the more resilient L3 stage. A testable range was then determined at four points, including a maximum dose (corresponding to each drug’s C_max_), a high dose (½ C_max_), a medium dose (¼ C_max_), and a low dose (1/10 C_max_). This range aimed to reflect decreasing availability of the drug after reaching peak plasma levels. The dose selection process is illustrated in Table 1.

#### 2.2.2. Preparation of Stock Solutions

Stock solutions of pyrantel, ivermectin, and emodepside were prepared in appropriate solvent mixtures of DMSO and BU buffer (50 mM Na_2_HPO_4_, 22 mM KH_2_PO_4_, 70 mM NaCl, pH 6.8) [26]. To account for solubility differences, pyrantel was prepared in 20% DMSO, ivermectin in 4% DMSO, and emodepside in 25% DMSO. Penitrem A was dissolved in 100% DMSO to make a 500 µM stock solution. A vehicle control solution was also prepared for each drug, containing the maximum DMSO concentration used in the stock solutions to ensure that solvent effects across treatments were adequately controlled.

### 2.3. Experimental Design of Survival Assays

#### 2.3.1. Setup and Assessment of Survival Assays: *A. caninum* L3

Infective *A. caninum* L3 were collected from coproculture at least 7 days post-culture using a modified Baermann method and stored in BU in a 50 mL culture flask at room temperature for up to four weeks prior to use. The survival assay to evaluate the response of *A. caninum* L3 to various anthelmintics was adapted from a previously established motility assessment study [27].

Prior to use, L3 were surface sterilized by incubation in 1% (*v*/*v*) HCl for 30 min, followed by two washes with sterile BU buffer. The sterilized larvae were resuspended in BU to a concentration of 75 L3 per 90 µL. Larvae were transferred in aliquots of 90 µL to individual wells in a 96-well plate, and 10 µL of the appropriate drug or control solution was added to each well, bringing the final volume to 100 µL. Plates were incubated at room temperature in the dark for 24 h.

Following incubation, survival was assessed by adding 50 microliters of 50° C water to each well immediately before counting [28]. Larvae were recorded as alive if they exhibited any consistent movement or as dead if they displayed the characteristic straightened appearance [26]. For each well, the total number of living and dead L3 were counted, and percent survival was calculated as below:(1)$$Survival\;(\%)=\frac{Dead\;L3}{Dead\;L3+Live\;L3}\times 100$$

#### 2.3.2. Evaluation of SLO-1 Modulation Using Penitrem A

To further investigate the specific mechanism of emodepside action, the SLO-1 channel inhibitor penitrem A was co-incubated with emodepside. The concentration of emodepside was kept constant (7.5 µg/mL) while that of penitrem A was titrated from 1 µM to 0.05 µM. A vehicle control well of 2.5% DMSO and 1 µM penitrem A was also included. *A. caninum* L3 were prepared identically to the previous drug survival assays. After addition to the assay, the larvae were incubated in the dark at room temperature for 24 h. Survival was assessed as above.

### 2.4. Data Collection and Analysis

Microsoft Excel was used to normalize raw survival percentages to their respective controls and calculate standard deviation for each condition. Statistical analysis was performed using GraphPad Prism (ver. 10, GraphPad Software, La Jolla, CA, USA). Using the mean, standard deviation, and sample size calculated in Excel, multiple unpaired *t*-tests were performed to determine the individual variance in means between WMD and BCR.

## 3. Results

### 3.1. A. Caninum L3 Survival Assays

In vitro larval survival assays were used to determine the efficacy of three anthelmintics (pyrantel, ivermectin, and emodepside) on *A. caninum* L3.

Across the tested concentrations, pyrantel was ineffective at killing *A. caninum* L3. At the lower concentrations, 10–50 µg/mL, almost no mortality was observed. Only at the highest dose of 100 µg/mL was reduced survival observed (Figure 1A). At this concentration, a slight difference between the susceptible WMD $\left(\bar{x}=43.9\pm 13.7\right)$ and resistant BCR $\left(\bar{x}=59.2\pm 10.4\right)$ isolates was detected; however, this difference was not significant (*p* = 0.054).

Ivermectin exhibited a strong lethal effect at each concentration tested. Even at the low dose (4 µg/mL), almost all WMD larvae were killed $(\bar{x}=14.9\pm 9.7$ normalized survival). This drug was successful at killing some BCR larvae $\left(\bar{x}=54.2\pm 11.4\right)$; however, the isolate showed marked resistance to ivermectin across all concentrations when compared with WMD. A two-tailed unpaired *t*-test revealed the difference in mean survival between BCR and WMD to be 39.0 ± 3.0 (*p* < 0.0001). For the concentrations tested, the L3 response to ivermectin was not dose-dependent. Notably, increasing the concentration from 4 to 40 µg/mL did not cause a significant change in percent survival for either isolate (Figure 1B). However, it should be noted that this effect may be due to the limited aqueous solubility of ivermectin (<10 µg/mL) rather than the drug reaching a potential activity maximum.

Emodepside was highly successful at killing *A. caninum* L3 across a majority of the tested concentrations (Figure 1C). Beginning at 3 µg/mL (low dose), the majority of larvae in both isolates were killed. Unexpectedly, a reversal in resistance trends became evident across all concentrations. Whereas emodepside killed all BCR larvae starting at 7.5 µg/mL, mean WMD survival was 13.1% ± 4.1 at the same concentration. Even up to 30 µg/mL, the maximum dose tested, 3.4% ± 3.3 of WMD larvae survived, versus none of the BCR larvae. The apparent higher susceptibility of the MADR isolate to emodepside was significant at almost all concentrations, with the greatest difference in percent survival appearing at 3 µg/mL $\left(\bar{x}=21.8\pm 4.4\right)$. The L3 response to emodepside appeared to be dose-dependent, with survival drastically decreasing at higher concentrations. To quantify the difference in emodepside sensitivity between isolates, a nonlinear regression was performed to generate dose-response curves for WMD and BCR. The resulting curves demonstrated a clear leftward shift for BCR, consistent with a lower IC_50_ value of 1.77 µg/mL, compared to 2.81 µg/mL for WMD (Figure 1D).

### 3.2. Emodepside + Penitrem A Survival Assays

To determine the mechanism underlying emodepside’s action on SLO-1/BK channels in *A. caninum* L3, larvae were co-incubated with emodepside and penitrem A, a SLO-1 channel inhibitor.

#### 3.2.1. Emodepside + 1 µM Penitrem A

As shown in Figure 2A, *A. caninum* L3 were killed in a concentration-dependent manner, with WMD L3 less susceptible to the drug, as seen previously. However, the addition of 1 µM penitrem A completely abrogated the lethal effect of emodepside at all tested concentrations. Between 90 and 98% of L3 from both isolates survived, even at 7.5 µg/mL, a concentration which was highly lethal in the absence of the inhibitor. Notably, the greater resistance of WMD L3 to emodepside disappeared in the presence of penitrem A (Figure 2A).

#### 3.2.2. Penitrem A Titration

Between 1 and 0.75 µM, penitrem A protected WMD and BCR equally from the lethal effects of emodepside (Figure 2B). Beginning at 0.5 µM and continuing down to 0.1 µM, survival was significantly higher (*p* = 0.005) for BCR than WMD. At 0.25 µM, BCR exhibited 78.2% ± 13.8 survival, with WMD survival significantly lower, at 29.8% ± 11.5. Higher survival in the resistant isolate remained consistent until 0.05 µM, the lowest tested concentration of penitrem A.

## 4. Discussion

In this study, the in vitro efficacy of emodepside, pyrantel, and ivermectin was tested against drug-susceptible (WMD) and multidrug-resistant (BCR) *A. caninum* L3. Considering BCR’s known resistance to pyrantel and ivermectin, we initially sought to confirm the utility of our in vitro survival assay by showing reduced efficacy of these drugs against BCR. Surprisingly, we found emodepside to be slightly more effective on the resistant isolate when compared to drug-susceptible WMD. Next, the mechanism of action of emodepside’s target, SLO-1, was investigated. Using the potassium channel antagonist penitrem A, we identified key phenotypic distinctions between WMD and BCR L3.

Our results show that the in vitro survival assay is uninformative as a measure of pyrantel efficacy for *A. caninum*. Although pyrantel is a well-established treatment for hookworms in vivo, we and others have found that its activity against free-living stages in vitro is limited [7,29], possibly due to life-stage-specific factors. Studies that have examined pyrantel’s activity against *A. ceylanicum* and *N. americanus* have noted significant discrepancies in efficacy that exist not only between species [30] but also between L3 and adults of the same species [31]. Such variability could feasibly explain the lack of activity for pyrantel observed in our L3 survival assay and highlight the difficulties in identifying pyrantel-resistant hookworm isolates via in vitro assays. By comparison, ivermectin was highly lethal against L3, and BCR exhibited approximately 40% greater survival than WMD across all tested concentrations. These results provided a baseline for comparison against emodepside as well as confirmation for the utility of the L3 survival assay as a means of evaluating ivermectin efficacy in vitro.

We next assessed the in vitro activity of emodepside, which had not previously been evaluated against either isolate. Interestingly, our triple-resistant isolate was slightly more susceptible to emodepside than our wild-type isolate, suggesting that the mechanisms that confer BCR’s resistance to other drugs do not protect—and may even sensitize—the larvae to emodepside. From a clinical perspective, the identification of a drug that not only circumvents existing resistance but also appears more potent in the resistant isolate is highly promising.

Emodepside exerts its lethal effect by targeting SLO-1, a calcium-gated potassium channel (BK channel) [32]. During emodepside exposure, the drug acts agonistically on these channels, causing a massive potassium efflux and inducing irreversible neuronal hyperpolarization [33]. Prolonged hyperpolarization prevents the firing of action potentials, resulting in paralysis, reduced pharyngeal pumping, and ultimately, death. Confirming whether BCR shows increased activity of this ion channel could help explain the increased sensitivity of BCR to emodepside. To explore whether BK channel differences underlie this phenomenon, we tested penitrem A, a BK channel antagonist, for its effect on emodepside activity. Theorizing that the effect of a potent antagonist (penitrem A) would counter the effect of the agonist (emodepside), we predicted that a low concentration of penitrem A should effectively prevent L3 death from emodepside. As expected, a low concentration of penitrem A rescued both isolates from emodepside-induced mortality at higher concentrations of emodepside, confirming that BK channel activity underpins emodepside’s lethal effect in hookworms. More notably, titrating penitrem A revealed a complete reversal of the previous trend: BCR became less susceptible than WMD. This outcome strongly suggests that the BK channels (SLO-1) of the resistant BCR isolate respond differently to modulation, potentially reflecting differences in channel density, conformation, or regulation.

It is important to note some key limiting features of our study, which include the reliance on in vitro assays and the need for validation in vivo. While essential for initial screening, the in vitro assays described here may not capture the full complexities of infection in vivo. Future in vivo studies involving fecal egg count reduction (FECR) tests in dogs infected with these specific isolates would confirm whether emodepside’s enhanced activity against BCR extends to adult worms and improves clinical outcomes. Furthermore, although the penitrem A assays suggest a mechanistic explanation involving BK channel activity, direct protein-level assays or electrophysiological measurements would strengthen the link between BK channel function and the influence of emodepside.

Because there is no standardized method for determining lethal in vitro drug concentrations for *A. caninum* L3, we developed a rationale based on known pharmacokinetic data. To do this, we obtained the maximum plasma concentration (C_max_) of each drug in vivo from the literature and multiplied this value by a factor of 1000 to approximate a testable concentration maximum that might better target larval-stage worms, which are generally less susceptible to the lethal effects of anthelmintics. This multiplier was selected based on preliminary observations that L3 often require significantly higher concentrations for observable effects in vitro [31]. The corresponding concentration range for each drug was then established by taking fractional doses of the maximum. Our results suggest that this method provides a reproducible framework for future studies targeting non-adult parasite stages and helps address the need for assay development specific to the infective larval stage.

This study contributes to the growing evidence that emodepside is a powerful alternative to the currently available treatments for hookworm infections, especially those involving multidrug-resistant isolates. Moreover, our results highlight a potential weakness in the BCR isolate that could be exploited: an increased sensitivity to a BK channel agonist. This discovery not only strengthens emodepside’s profile as a treatment for drug-resistant infections but also informs the development of novel treatments seeking to target similar vulnerabilities. An increased focus on these factors could lead to the development of novel anthelmintics or combination therapies that remain effective against otherwise resistant parasites.

While further in vivo validation is warranted, our findings underscore the promise of emodepside for managing multidrug-resistant *A. caninum* infections. By elucidating how BK channel modulation influences larvae survival, we open new avenues for research into the molecular basis of resistance and highlight opportunities to develop more effective anthelmintic strategies.

## 5. Conclusions

Emodepside, a worthy candidate for the treatment of resistant hookworm infections, uses a unique mode of action to execute its lethal effect, targeting the parasite’s SLO-1 (BK channel) receptors. This study employed a simple L3 survival assay to test the effect of emodepside on wild-type (WMD) and resistant (BCR) *A. caninum* L3 compared with two more commonly used anthelmintics, pyrantel and ivermectin. Pyrantel proved to be ineffective against L3, while ivermectin produced significant mortality. The success of ivermectin in the L3 survival assay allowed us to reliably investigate emodepside using this method. The research presented here demonstrates the strength of emodepside against WMD, but more notably, the superior efficacy against BCR. The enhanced action of emodepside against BCR led us to postulate about the sensitivity and function of BK channels in the resistant isolate. This hypothesis was confirmed with the evaluation of a BK channel inhibitor, penitrem A, which showed the ability to rescue BCR L3 from the lethal effect of emodepside when the two were co-incubated. This study not only bolsters the efficacy profile of emodepside, proving its merit as a treatment method for hookworms, but also reveals an important and distinct characteristic of this resistant isolate, providing novel data that may help shape our understanding of MADR parasites and how best to treat their persistent infections.

## Figures and Tables

**Figure 1 tropicalmed-10-00181-f001:**
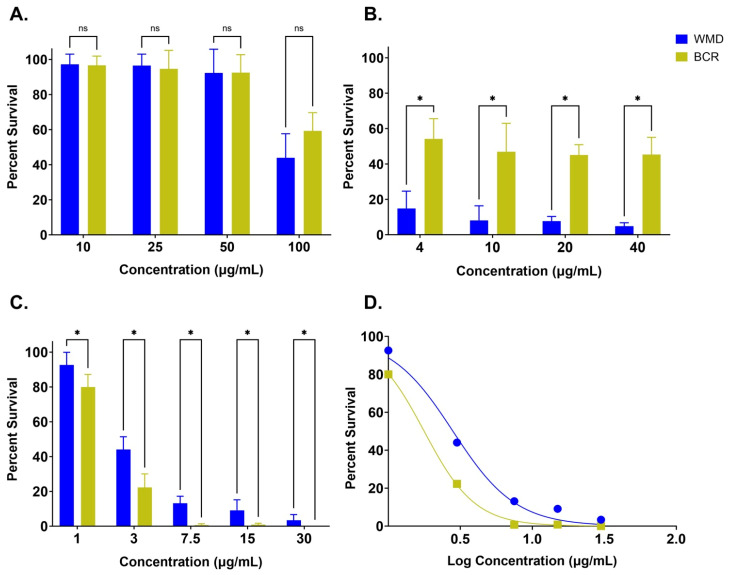
In vitro survival of drug-susceptible WMD (blue) and drug-resistant BCR (yellow) *A. caninum* L3 after exposure to three anthelmintics and corresponding emodepside dose curve. Percent survival of WMD and BCR larvae at increasing concentrations of (**A**) pyrantel, (**B**) ivermectin, and (**C**) emodepside. (**D**) A nonlinear regression curve plotting log-transformed emodepside concentrations against percent survival highlights the lower IC_50_ in BCR (1.77 µg/mL) when compared to WMD (2.81 µg/mL). Percent survival for all data points A-C is normalized to the respective vehicle control for each drug. Statistical analysis was performed using multiple unpaired *t*-tests, * *p* < 0.05. The notation “ns” is used to mark data points where no significant difference in survival was observed.

**Figure 2 tropicalmed-10-00181-f002:**
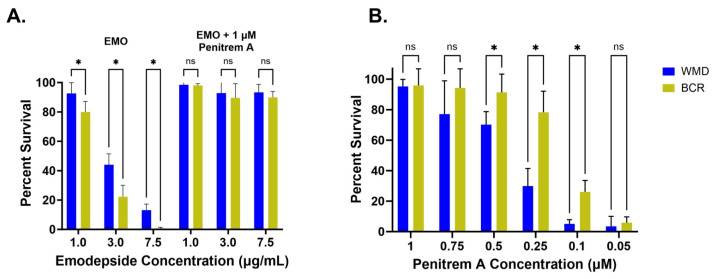
In vitro survival of drug-susceptible WMD (blue) and drug-resistant BCR (yellow) *A. caninum* L3 exposed to emodepside alone or emodepside and penitrem A. (**A**) 1 µM penitrem A was used for all conditions, whereas emodepside ranged between 1 and 7.5 µg/mL. (**B**) L3 were exposed to a lethal concentration (7.5 µg/mL) of emodepside and a range of concentrations (1–0.05 µM) of penitrem A. Percent survival for all data points is normalized to the vehicle control, 2.7% DMSO. Statistical analysis was performed using multiple unpaired *t*-tests, * *p* < 0.05.

**Table 1 tropicalmed-10-00181-t001:** Selection of testable concentration ranges for pyrantel, ivermectin, and emodepside ^1^.

Drug	Administered Dose	Corresponding C_max_ (µg/mL)	X1000 (µg/mL)	Testable Range (µg/mL)
Pyrantel	100 mg/kg	0.11	110	10, 25, 50, 100
Ivermectin	150–200 µg/kg	0.03–0.04	30–40	4, 10, 20, 40
Emodepside	2.5 mg	0.037	37.6	1, 3, 7.5, 15, 30

^1^ Drug concentrations were based on typical oral dose and corresponding maximum plasma concentration (C_max_) values, which were obtained from the literature. To target the more resilient *A. caninum* L3 stage, C_max_ values were multiplied by a factor of 1000 to create a high-concentration value. From this value, four concentrations were selected for each drug: the maximum dose (C_max_), high dose (1/2 C_max_), medium dose (1/4 C_max_), and low dose (1/10 C_max_). One additional concentration equal to 1/30 C_max_ was tested for emodepside.

## Data Availability

The raw data supporting the conclusions of this article will be made available by the authors upon request.

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
