# Peer review of "Differential Emodepside Efficacy in Drug-Resistant and Drug-Susceptible Ancylostoma caninum Highlights Variability in Potassium Channel Activity"

_tropicalmed, 2025, doi:10.3390/tropicalmed10070181_

Round 1
Reviewer 1 Report
Comments and Suggestions for Authors
This is an interesting and important manuscript that address an issue of current concern in veterinary parasitology. I have only minor suggestions for improvement.
Line 39: the authors should note that CLM is the manifestation of A. canine infection in humans to be precise, and that the drugs used to treat it in humans are the same as those affected in MADR parasites, generating a therapeutic challenge; in the same area, the authors should define triple drug resistance by presenting pyrantel.
Line 40: around the globe? As far as I know MADR strains remain a concern in mainland USA and perhaps Canada.
Line 43: Mycelia incognita was the default name of the PF1022A producer (fungi are identified on the basis of mycelia and this one didn't initially make mycelia); it is now known to be produced by a Rosellina spp.
Line 51-54: Italicize genus and species; this is a bit of a concern throughout the text; also pay attention to not using the genus name after first use unless it introduces a sentence.
The authors need to note that emodepside is licensed for use in cats and dogs in much of the world and in cats in North America; it has broad-spectrum activity against GI nematodes in many host species, including companion animals.
Line 80: probably mean plasma, not serum
Line 88: The recent paper describing the PK of emodepside in the approved oral form for dogs in the EU and the cat topical formulation administered orally (Quintana et al.) may deserve a bit more text here.
Line 110: attend to subscripts (including IC50)
Line 162: “A slightly decreased survival for BCR was evident (xÌ„ _= 54.2 ± 11.4), although BCR 162 showed marked resistance to IVM across all concentrations.” This is unclear to me and seems contradictory. Please clarify.
Line 166: IVM aqueous solubility maximum is <10 ug/ml, so the lack of concentration-dependent effects is unsurprising.
Author Response
The authors would like to thank the reviewer for their careful consideration and detailed evaluation of this manuscript. Please find our responses to each of your comments below:
Comment 1: "Line 39: the authors should note that CLM is the manifestation of A. canine infection in humans to be precise, and that the drugs used to treat it in humans are the same as those affected in MADR parasites, generating a therapeutic challenge; in the same area, the authors should define triple drug resistance by presenting pyrantel."
Response 1: Thank you for addressing this point, we agree that this section of the introduction lacked clarification about the topic. To clear this up, we have added information about CLM in humans (lines 39-41), included pyrantel in the description of common treatments (line 36), and expanded on why this represents a challenge in both human and veterinary medicine (lines 43-45).
Comment 2: "Line 40: around the globe? As far as I know MADR strains remain a concern in mainland USA and perhaps Canada."
Response 2: Thank you for pointing this out-- you are correct. I had mistakenly generalized to MADR in human hookworm, although previous studies have highlighted that a lack of evidence does not necessarily negate the possibility of existence. We have added some information (lines 46-48) that hopefully will clear up any confusion on where MADR hookworm is believed to be prevalent.
Comment 3: "Line 43: Mycelia incognita was the default name of the PF1022A producer (fungi are identified on the basis of mycelia and this one didn't initially make mycelia); it is now known to be produced by a Rosellina spp."
Response 3: Thank you for pointing this out. We have corrected the information and included the relevant sources to reflect this update (line 51).
Comment 4: "Line 51-54: Italicize genus and species; this is a bit of a concern throughout the text; also pay attention to not using the genus name after first use unless it introduces a sentence."
Response 4: Thank you for highlighting this mistake. All instances throughout the text have been corrected. We believe this may have occurred due to a conversion issue when uploading the manuscript, and will be communicating with the editors to ensure that it does not happen again.
Comment 5: "The authors need to note that emodepside is licensed for use in cats and dogs in much of the world and in cats in North America; it has broad-spectrum activity against GI nematodes in many host species, including companion animals."
Response 5: We agree that this is an important note to address. We have added information that details the current availability of emodepside for veterinary use (lines 67-69), although we believe that the preceding paragraph is sufficient to accurately express the broad-spectrum efficacy of emodepside against a range of parasitic nematodes in a variety of host species.
Comment 6: "Line 80: probably mean plasma, not serum"
Response 6: Thank you for indicating this mistake-- all instances throughout the text have been corrected.
Comment 7: "Line 88: The recent paper describing the PK of emodepside in the approved oral form for dogs in the EU and the cat topical formulation administered orally (Quintana et al.) may deserve a bit more text here."
Response 7: The authors agree that this section was lacking a comprehensive account of the Quintana et al. study. We have added more to this section (lines 99-101), hopefully adding clarity to our reasoning behind the chosen in vitro concentrations for this drug.
Comment 8: "Line 110: attend to subscripts (including IC50)"
Response 8: Thank you for noticing this mistake. All subscripts have been corrected.
Comment 9: "Line 162: “A slightly decreased survival for BCR was evident (xÌ„ _= 54.2 ± 11.4), although BCR 162 showed marked resistance to IVM across all concentrations.” This is unclear to me and seems contradictory. Please clarify."
Response 9: We agree that the wording of this sentence is contradictory and confusing. Our intention was to point out that ivermectin is capable of killing A. caninum at the L3 stage where pyrantel was not, but the drug is still far less effective against the resistant isolate. We have revised the sentence so that hopefully our intended point is now more clear (lines 173-175).
Comment 10: "Line 166: IVM aqueous solubility maximum is <10 ug/ml, so the lack of concentration-dependent effects is unsurprising."
Response 10: Thank you for bringing this up-- we did run into solubility issues with this assay, and this is an important element to mention. We have included a note about this in the results section so that readers will be aware of this (lines 179-180).
Thank you again for taking the time for a precise review of this manuscript. The authors are very grateful for your contributions to this submission.
Reviewer 2 Report
Comments and Suggestions for Authors
This manuscript evaluated the in vitro efficacy of three anthelmintics namely pyrantel, ivermectin, and emodepside against infective larvae of drug-susceptible (WMD) and triple-anthelmintic-resistant (BCR) isolates (previously described by the research team) of Ancylostoma caninum. The findings revealed superior efficacy by emodepside compared to other drugs; with a higher efficacy observed on the BCR than the WMD isolates. On the other hand pyrantel was found ineffective while ivermectin showed high efficacy in both isolate types.
A key limitation of the study is the lack of in vivo validation of the results. However, this limitation was clearly acknowledged by the authors, with a recommendation for future studies to include in vivo validation.
Overall, the manuscript is well-written and presents important results that would be useful in the global efforts against soil-transmitted helminths including hookworms. I found the manuscript acceptable for publication; however, some minor typographic errors and minor comments are outlined below for the authors' consideration.
MINOR COMMENTS
- Introduction: The first two sentences (lines 26-30) are general information on the global burden of human hookworm infections (by N.a. & A.d.). The 4th sentence (lines 33-36) mentioned the spread of MARD hookworm due to widespread and sometimes improper use of the indicated drugs (albendazole & ivermectin), with 4 references were cited (Ref. 2-5). All those references (i.e. #2-5) were on canid hookworms which appeared for the first time in the next sentence (line 36). Thus, it is useful to add (after the 2nd sentence) a sentence on the burden or distribution of animal hookworms or at least the canid ones. Also, add references on resistance in human hookworms, if applicable.
- Lines 52-53: Scientific names such as Trichuris muris, Ancylostoma ceylanicum, Necator americanus, Heligmosomoides polygyrus, and Strongyloides ratti, as well as their abbreviated forms ( muris, N. americanus, A. ceylanicum), should be italicized. Also, in vitro and in vivo in the same sentence should be italicized. Please apply these consistently throughout the manuscript (e.g., lines 57, 154, 155, etc.).
- Line 75: add the abbreviation “ …..and dimethyl sulfoxide (DMSO)….
- Line 101: correct to “Table 1.”. All legends following the table titles should be moved to the table footnotes.
- Table 1: Capitalization should be standardized—for example, "Testable Range" and "Administered Dose." Please apply consistent capitalization throughout the manuscript, including in subsection titles such as "2.4. Data Collection and Analysis," where applicable.
- Line 163: Please replace "IVM" with "ivermectin," as the abbreviation appears only once and is unnecessary.
Author Response
The authors would like to thank the reviewer for their careful consideration and detailed evaluation of this manuscript. Please find our responses to each of your comments below:
Comment 1: "Introduction: The first two sentences (lines 26-30) are general information on the global burden of human hookworm infections (by N.a. & A.d.). The 4th sentence (lines 33-36) mentioned the spread of MARD hookworm due to widespread and sometimes improper use of the indicated drugs (albendazole & ivermectin), with 4 references were cited (Ref. 2-5). All those references (i.e. #2-5) were on canid hookworms which appeared for the first time in the next sentence (line 36). Thus, it is useful to add (after the 2nd sentence) a sentence on the burden or distribution of animal hookworms or at least the canid ones. Also, add references on resistance in human hookworms, if applicable."
Response 1: Thank you for pointing this out. We agree that the wording and order of topics introduced here may be misleading for the overall topic of the paper. We have added a sentence with information about A. caninum before the introduction of common drug treatments to create a better flow. We have not included any references on resistance in human hookworms to avoid confusion about the applicability of the research.
Comment 2: "Lines 52-53: Scientific names such as Trichuris muris, Ancylostoma ceylanicum, Necator americanus, Heligmosomoides polygyrus, and Strongyloides ratti, as well as their abbreviated forms ( muris, N. americanus, A. ceylanicum), should be italicized. Also, in vitro and in vivo in the same sentence should be italicized. Please apply these consistently throughout the manuscript (e.g., lines 57, 154, 155, etc.)."
Response 2: Thank you for highlighting this mistake. All instances throughout the text have been corrected. We believe this may have occurred due to a conversion issue when uploading the manuscript, and will be communicating with the editors to ensure that it does not happen again.
Comment 3: "Line 75: add the abbreviation “ …..and dimethyl sulfoxide (DMSO)…."
Response: This abbreviation has now been included.
Comment 4: "Line 101: correct to “Table 1.”. All legends following the table titles should be moved to the table footnotes."
Response 4: The table 1 caption has been corrected and legend has been moved to the footnotes.
Comment 5: "Table 1: Capitalization should be standardized—for example, "Testable Range" and "Administered Dose." Please apply consistent capitalization throughout the manuscript, including in subsection titles such as "2.4. Data Collection and Analysis," where applicable."
Response 5: "We have corrected all instances of incongruent capitalization in the table as well as throughout the text.
Comment 6: "Line 163: Please replace "IVM" with "ivermectin," as the abbreviation appears only once and is unnecessary."
Response 6: This abbreviation has been removed.
Thank you again for taking the time for a precise review of this manuscript. The authors are very grateful for your contributions to this submission.